# Benzothiadiazole Affects Grape Polyphenol Metabolism and Wine Quality in Two Greek Cultivars: Effects during Ripening Period over Two Years

**DOI:** 10.3390/plants12051179

**Published:** 2023-03-04

**Authors:** Dimitrios-Evangelos Miliordos, Anastasios Alatzas, Nikolaos Kontoudakis, Marianne Unlubayir, Polydefkis Hatzopoulos, Arnaud Lanoue, Yorgos Kotseridis

**Affiliations:** 1Laboratory of Oenology and Alcoholic Beverage Drinks, Department of Food Science and Human Nutrition, Agricultural University of Athens, 75 Iera Odos, 11855 Athens, Greece; 2Molecular Biology Laboratory, Department of Biotechnology, Agricultural University of Athens, 75 Iera Odos, 11855 Athens, Greece; 3EA 2106 Biomolécules et Biotechnologie Végétales, UFR des Sciences Pharmaceutiques, Université de Tours, 31 Av. Monge, F37200 Tours, France; 4Department of Agricultural Biotechnology and Oenology, International Hellenic University, 1st Km Drama-Mikrochori, 66100 Drama, Greece

**Keywords:** *Vitis vinifera* L., Savvatiano, Mouhtaro, biostimulants, benzothiadiazole, polyphenolic profile, gene expression, phenylpropanoid biosynthesis

## Abstract

Grape berries are one of the most important sources of phenolic compounds, either consumed fresh or as wine. A pioneer practice aiming to enrich grape phenolic content has been developed based on the application of biostimulants such as agrochemicals initially designed to induce resistance against plant pathogens. A field experiment was conducted in two growing seasons (2019–2020) to investigate the effect of benzothiadiazole on polyphenol biosynthesis during grape ripening in Mouhtaro (red-colored) and Savvatiano (white-colored) varieties. Grapevines were treated at the stage of veraison with 0.3 mM and 0.6 mM benzothiadiazole. The phenolic content of grapes, as well as the expression level of genes involved in the phenylpropanoid pathway were evaluated and showed an induction of genes specifically engaged in anthocyanins and stilbenoids biosynthesis. Experimental wines deriving from benzothiadiazole-treated grapes exhibited increased amounts of phenolic compounds in both varietal wines, as well as an enhancement in anthocyanin content of Mouhtaro wines. Taken together, benzothiadiazole can be utilized to induce the biosynthesis of secondary metabolites with oenological interest and to improve the quality characteristics of grapes produced under organic conditions.

## 1. Introduction

Grapevine (*Vitis vinifera* L.) is a perennial woody plant with a prolonged cultivation history in Europe since the second millennium B.C., revealing a fundamental socioeconomic impact [1,2,3]. The Greek vineyard represents approximately an area of 60,000 ha, planted mainly (i.e., over 90%) with indigenous varieties such as Agiorgitiko, Xinomavro, Savvatiano, and Assyrtiko [4]. Savvatiano is an autochthonous Greek, white-colored variety representing more than 16% of the Greek vineyard, and Mouhtaro, a rare autochthonous red-colored variety, is planted mostly at the Muses Valley in the area Viotia (Central Greece), representing more than 90% of the local vineyard [4]. 

Sustainable viticulture, including organic production, is typically characterized by a lower crop yield compared to conventional production systems, mainly due to the limitation imposed on fertilization (no use of chemical fertilizers) and on plant protection and pathogens (no use of pesticides) [5,6]. Modern agriculture tends to minimize the use of chemical plant protection agents and to replace them with compounds of natural origin [7]. This group of compounds includes biostimulants that are used to alleviate environmental stress [8]. In the last decade, research groups around the word study the application of biostimulants on different grapevine varieties [9,10,11,12]. Despite the large and increasing number of publications dealing with biostimulants [13], scientific-based information on their optimal use and crop specificity relative to the growing conditions is still incomplete.

Phenolic compounds represent a class of intensively studied bioactive molecules which are related with the organoleptic characteristics of wines, such as color, astringency and bitterness, as well as wine stability through enzymatic and chemical oxidative mechanisms. Moreover, they possess numerous health benefits and are used in the pharmaceutical industry for the treatment of various diseases [14,15,16,17]. They are also utilized in the food industry as additives, natural preservatives, and dyes as well as in the cosmetic industry due to its antimicrobial, antioxidant, and anti-inflammatory properties [18,19]. Up to date, there are about 5000 phenolic compounds of plant origin, and they vary in response to plant species and environmental conditions [20]. 

The content and profile of polyphenols in grape are affected by the cultivar, environmental factors, and viticultural practices. 

In the last 10 years, several studies have been conducted regarding the impact of biostimulants. The exogenous applications of biostimulants induce the activation of enzymes involved in the biosynthesis of phenolic compounds [21,22,23]. BTH is considered to be an analogue of Salicilic Acid inducing resistance against a broad spectrum of plant pathogens [24,25]. However, few research studies have focused on the impact of benzothiadiazole (BTH) on the quality of grapes and wines. For instance, the contents of anthocyanin, flavonols, stilbenes, and tannins of grapes were increased after the BTH application of various clones of the Monastrell variety [26]. Benzothiadiazole also increased the levels of anthocyanins [27]. An increase in aromatic compounds of wines after the application of BTH was also recorded by Gómez-Plaza et al. [28] and by Vitallini et al. [29]. However, the BTH application on the biosynthesis of phenolic compounds showed that it is variety-dependent, as shown in Monastrell, Merlot, and Syrah varieties [30].

Herein, we evaluated the application of biostimulant BTH at a pre-harvest stage and determined the increased contents of phenolic compounds in grapes and wine. Changes in the profile of phenolic compounds and gene expression upon BTH treatment were determined during ripening period in the red-colored variety Mouhtaro and the white-colored Savvatiano. The produced wines were analyzed and the extractability of phenolic compounds during winemaking was evaluated showing an enhancement of phenolic content.

## 2. Results

### 2.1. Meteorological Data for the Two Years of Study

Savvatiano is characterized as a late-harvested variety according its phenological stages at the Muses Valley. In the two vintages of the trial, budbreak occurred on 20 and 14th March in 2019 and 2020, respectively, while veraison started around 15th August and 20th September in 2019 and 2020, respectively. The temperatures recorded between the two experimental years showed no significant differences, whilst the total amount of rainfall was higher during the 2020 season (Table 1).

Mouhtaro is an earlier-harvested variety compared to Savvatiano. Budbreak occurred on 8th and 3rd March in 2019 and 2020, respectively, and veraison started on 9th August and 15th July in 2019 and 2020, respectively. According to the meteorological data, August was drier in 2019 compared to 2020. In addition to the higher rainfall recorded in 2020, the mean temperature at the post-veraison stage was higher than 2019. Therefore, the 2020 ripening period was longer than 2019 for the Mouhtaro variety.

### 2.2. Physicochemical Parameters of Savvatianno Grape Berries

The high dose of BTH treatment exhibited significantly lower berry size values at all phenological stages of both vintages (Table 2). The higher berry weight values observed in 2020 vintage could be the result of higher water absorption by the plant due to the higher amount of rainfall recorded in 2020 (Table 1). A similar effect of weather conditions has been also described in the Monastrell variety by Paladines Quezada et al. [31]. 

Total soluble solids were increased throughout grape development (Table 2). The control samples recorded slightly lower ^o^Brix values than the BTH-treated samples (low and high doses) in 2019 vintage. The differences were not substantial in 2020 vintage, except at the harvest stage. Control samples showed significantly higher ^o^Brix values (20.13 ^o^Brix) compared to both high (19.10 ^o^Brix) and low (19.53 ^o^Brix) BTH treatments. Grapes treated with BTH showed lower sugar concentration than the control grapes, as observed in 2020, indicating that BTH treatment might delay the ripening process of Savvatiano variety. Similar results were also recorded after the exogenous application of the biostimulants chitosan and abscisic acid on Savvatiano grapes during 2020 vintage [32]. Previous studies concerning BTH application showed that it can delay fruit-ripening [33,34]. The pH content in all treatments was increased gradually, with no statistical differences among treatments (Table 2). However, grapes treated with the high BTH dose recorded the highest pH value (3.39) among all treatments in both vintages. On the other hand, titratable acidity (TA) values were decreased during the ripening period and no significant differences were observed among treated and control plants (Table 2). Our results concerning the physiochemical parameters are in agreement with the findings of Ruiz-García et al. [35], who recorded increased berry weight upon BTH application in Monastrell grapes in one of the two seasons.

### 2.3. Metabolomic Analysis of Savvatiano Grape Berries in Response to BTH

In the last decade, several studies have demonstrated that the application of benzothiadiazole modulates secondary metabolism [21,27,31,36]. However, the impact of BTH on white-colored grape varieties is unknown. This prompted us to perform a targeted metabolomic analysis and evaluate the changes in specific secondary metabolites in response to different concentrations of BTH in the white-colored variety, Savvatiano. During the 2020 vintage, the higher mean temperature during the post-veraison stage coupled with higher precipitation than 2019 (Table 1) resulted in grape berries with a higher level of weight/berry ratio. Consequently, the concentration of the phenolic compounds in berries was lower in 2020 vintage than in 2019 vintage (Figure 1 and Appendix A). In the two consecutive vintages, BTH did not substantially affect the majority of berry phenolic compounds (Figure 1 and Appendix A). 

Metabolomic analyses showed that different doses of BTH differentially affected the metabolic composition, including amino acids, stilbenoids, flavonols, flavan-3-ols, and anthocyanins diOH in Savvatiano grapes in all three phenological stages examined. Six amino acids were identified in all treatments during grape maturation, namely, L-proline, L-leucine, L-isoleucine, L-phenylalanine, L-tyrosine, and L-tryptophan. The amino acids exhibited higher levels in the treated grape berries than the controls in the two first phenological stages, presenting no statistical differences except the middle veraison stage of 2019 vintage (Appendix A). The treated grape berries showed slightly lower levels than the control ones in 2019 harvest, and the BTH high dose showed higher concentration levels of total amino acids than the BTH low dose and control grapes in 2020 harvest. However, no statistically significant differences were observed among the treatments in both vintages (Figure 1). 

The BTH-treated berries recorded significantly higher concentration levels of total anthocyanins diOH than the controls at all phenological stages of 2019 vintage (Figure 1 and Appendix A). Although the same trend was also observed in 2020 vintage, the differences were not statistically significant (Appendix A and Figure 1). Similar results concerning the total anthocynanins diOH in Savvatiano were also recorded after the application of the biostimulants of chitosan and abscisic acid, as shown by Miliordos et al. [32]. Conventional HPLC methods which are used in order to detect and quantify grape anthocyanin are not sensitive enough to detect pigments at the level of a few μg/kg grapes. However, modern UPLC-MS/MS instruments which are characterized by a higher number of chromatographic theoretical plates and a higher sensitivity detector (triple quadruple MS) are also able to detect and quantify traces of anthocyanins. So far, few research studies have demonstrated the existence of anthocyanins in white grapevine varieties, such as in *Vitis vinifera*, L. Siria. [37], Riesling, and Sauvignon Blanc [38].

An additional phenolic group detected by the UPLC–MS was the stilbenoids. Specifically, piceid, *E*-resveratrol, *E*-piceatannol, and *E*-*ε*-viniferin were detected in grape samples. The effect of the BTH application was negligible at the middle veraison stage of both vintages (Appendix A), while it was evident at the harvest stage (Figure 1). Notably, the high dose of the BTH significantly increased the total stilbenoids level in both vintages. Although the low dose BTH exhibited higher total stilbenoids than the control grapes, the difference was not statistically significant (Figure 1). An increased level of stilbenoids in Savvatiano at the harvest stage was also observed after the application of chitosan and abscisic acid, according to Miliordos et al. [32]. Stilbenoids are well-known grape phytoalexins that are induced following environmental stress, such as fungal infections [38,39]; however, the stilbenoid induction may also affect plant and berry development [40].

In this study, eight flavan-3-ols were detected by UPLC–MS, namely, catechin, epicatechin, catechin gallate, procyanidin B1, procyanidin B2, procyanidin B3, procyanidin B4, and procyanidin gallate. A consistent decrease in the total flavan-3-ols content in the grape berries over the ripening period was recorded, while the BTH treatments did lead to significant changes at veraison (Appendix A) and harvest stage (Figure 1). For instance, the BTH high dose significantly decreased the concentration of total flava-3-ols at the veraison stage in both vintages (Appendix A). On the other hand, the BTH low or high dose significantly increased the flava-3-ols concentration at the harvest stages of 2019 and 2020, respectively (Figure 1). 

Similarly, the flavonols level was decreased over the ripening period. BTH application did not induce the flavonol content at the veraison stage of the 2019 and 2020 vintage (Appendix A). Similar results were observed at the middle veraison stage, when the flavonol content was lower in the BTH-treated vines than the controls, while in 2019 vintage, the treated berries showed significantly lower flavonol content than control (Appendix A). Although differences between treated and control vines were recorded at harvest of both vintages, only the BTH low dose led to grapes with significantly higher flavonol content than controls at the harvest stage of 2019 vintage, while in 2020 vintage, no differences were observed (Appendix A). 

Four phenolic acids were detected by UPLC–MS, namely, gallic acid, coutaric acid, caftaric acid, and fertaric acid. A consistent decrease in the total phenolic acid concentration in grape berries was recorded over the ripening period, while BTH treatments did not lead to any significant change (Figure 1 and Appendix A). 

### 2.4. Physicochemical Parameters of Savvatiano Experimental Wines

All Savvatiano wines showed usual values of enological parameters according to the physiochemical grape berry data. In the second vintage (2020), wines presented significant differences among the control wines and wines made from treated grapevines (BTH low and high) regarding alcoholic degree, and both treatments produced wines with decreased alcoholic degrees. In contrast, no significant difference was observed among the treatments and the control wines in 2019 vintage. Furthermore, BTH treatments increased the total acidity of wines compared to the control wines, without any statistically significant difference (Table 3). Similarly, no significant difference was observed in pH values in wines produced from treated and control wines (Table 3). Similar results were obtained upon the application of ethephon to the white-colored variety, Verdejo [41]. Finally, the volatile acidity values were below 0.6 g/L, which is usually perceived as a spoilage character for wine at high concentrations. 

### 2.5. Color and Phenolic Parameters of the Three Experimental Savvatiano Wines in Response to Two Different Treatments and a Control

According to our results, BTH treatment (especially the high dose) significantly increased the phenolic parameters of wine, such as Total Phenolic Index, color (420 nm absorbance), and Total Polyphenol Concentration value, in 2019 vintage. On the contrary, the control wines provided a different profile in 2020 vintage compared to the 2019, recording higher levels of the wine phenolic parameters than the BTH wines (Table 4). Additionally, BTH-treated wines showed higher k factor levels than the control wines in both vintages, suggesting that BTH-treated wines could provide brown color earlier than the controls (Table 4). To our knowledge, no data have been presented so far concerning the phenolic profile of white wines produced from grapes treated with BTH or with any other bioelicitors.

### 2.6. Physicochemical Parameters of Mouhtaro Grape Berries

BTH treatment resulted in smaller grape berries, with higher sugar content and higher total acidity in 2020 vintage. Different climatic conditions could have influenced grape ripening and physicochemical composition. The weight/berry value (Table 5) gradually increased throughout ripening in both vintages and significant differences among BTH-treated and control grapes were observed. BTH-treated grapes were smaller in size than the control grapes in both vintages in all sampling dates. For instance, the observed berry weight of the control and BTH-treated (high dose) at harvest stage was 2.11 g and 2.01 g for the 2019, and 2.05 g and 1.96 g for 2020 vintage, respectively. It should be mentioned that berry size is considered a quality factor in the red varieties, as the skin is the area of the grape berry from which phenolic compounds will be released during the vinification process [42]. Hence, a smaller grape berry will result in a higher skin-to-pulp ratio, which means that the phenolic compounds will be less diluted in the grape must during winemaking [27]. However, despite the higher amount of rainfall during the 2020 ripening period, the ratio was smaller due to the higher temperature mean compared to 2019 vintage. Hence, the weather conditions could have led to a longer ripening period and smaller grape berries. 

The increasing trend of TSS in all treatments was similar to that of the weight. TSS (Table 5) in grape samples increased during the ripening period, with the control samples recording slightly higher ^o^Brix values than the BTH-treated samples. However, the differences were not substantial except at the harvest stage of 2020 vintage (Table 5). Similar results were observed in the Syrah variety by Fernandez-Marin et al. [43], while controversial results were recorded in the Monastrell variety [35]. BTH treatments produced grapes with a lower sugar concentration, as observed in the 2020 season, indicating that BTH treatment might delay the ripening process of the red variety, as in the Savvatiano variety.

Titratable acidity (TA) values decreased during the ripening period and TA in 2019 recorded lower values in BTH-treated berries compared to the controls. In contrast, the TA values of the treated berries showed higher values than the controls in 2020 vintage. On the other hand, pH content was increased gradually during maturation and no statistically significant differences were observed between treatments in the two first phenological stages (Table 5). However, the pH value of the BTH-treated berries recorded significantly higher values than the control ones at harvest of both vintages (Table 5). 

### 2.7. Metabolomic Analysis of Mouhtaro Grape Berries in Response to BTH Applications

The phenolic groups detected by UPLC-MS in Mouhtaro grape berries after the application of two different doses of benzothiadiazole in 2019 and 2020 vintages are presented in Figure 2 and in Appendix A). Amino acids represent the most important part of the total nitrogen content in grape must [44] since they are metabolized by yeast during the yeast growth phase and wine [45]. Total amino acids (L-proline, L-leucine, L-isoleucine, L-phenylalanine, L-tyrosine, and L-tryptophan) were found to decrease during the ripening period (Appendix A). At all phenological stages examined, BTH treatment (mainly high dose) increased the total amino acid concentration compared to the untreated vines (Appendix A), but the differences observed were not statistically significant, except at the middle veraison stage of both vintages (Appendix A). In contrast, lower concentrations of amino acids have been recorded upon BTH application in Cabernet Gernischt grapes [21].

Anthocyanin biosynthesis in Mouhtaro grapes during 2019 was higher than in 2020 (Figure 2 and Appendix A) at all phenological stages. Therefore, it can be assumed that environmental conditions could strongly affect the accumulation of anthocyanins in this variety, probably by inhibiting biosynthesis and/or degradation [46]. As mentioned above, the amount of rainfall was higher in the 2020 growing season compared to 2019 (Table 1), which could have affected the accumulation and distribution of anthocyanins in the skins of the grape berries. The decision of harvest was determined by the technological maturity, and not by the phenolic maturity, hence, it could be that the Mouhtaro grapes were not harvested at the point of their highest polyphenolic concentration. The application of BTH positively influenced the concentration of total anthocyanins in 2019 vintage but the differences recorded were not statistically significant (Appendix A and S4). On the other hand, BTH treatment increased anthocyanins diOH concentration at the harvest stage of 2020. The BTH low dose recorded significantly higher levels of anthocyanins in grapes than the controls (Appendix A). The positive effect of BTH on the content of anthocyanins has been also observed in Merlot and Monastrell varieties [26,47]. Anthocyanins triOH in untreated vines were constantly increased during the ripening period in both vintages. The application of BTH low dose resulted in significantly higher athocyanins triOH levels compared to the controls at the middle veraison stage of 2019 vintage. However, the same treatment tended to decrease the anthocyanins triOH content in 2020 vintage (Appendix A and Figure 3). 

Flavan-3-ols (especially procyanidin) are related to color stability through reactions of copigmentation and polymerization, and together with anthocyanins, they determine the ability of wines to age [48]. BTH treatment had a negative impact on the content of total flavan-3-ols in both vintages (Appendix A). However, both BTH treatments resulted in a significantly higher level of total flavan-3-oles than the controls at the harvest stage of 2020 (Figure 2). Remarkably, a wine with higher astringency and bitterness could be produced by grapes with the excess of these phenolic compounds [49], an important characteristic of fresh wines of high quality.

Flavonols are found in lower proportions in red-colored varieties since they form very stable copigmentation complexes with anthocyanins [50]. The higher temperatures recorded in 2020 (Table 1) combined with the greater number of days between veraison and harvest (54 days in 2020 vs. 49 in 2019, Table 1), could have increased flavonols level, compared to 2019. Koyama et al. [51] showed that biosynthesis of flavonols in grapes depends, to a greater extent than other phenolic compounds, on light exposure, which favors their accumulation in Cabernet sauvignon grapes by increasing the hours of solar radiation. BTH application resulted in a slightly higher level of flavonols at the harvest stage, but the difference was not statistically significant (Appendix A).

Four phenolic acids, namely, gallic acid, coutaric acid, fertaric acid, and caftaric acid were detected in Mouhtaro grape berries. BTH treatments did not lead to any significant difference in Mouhtaro grape berries except in the stage of middle veraison in 2019 vintage, and led to significantly decreased levels of phenolic acids (Appendix A). 

Regarding stilbenes, *E*-piceid was detected by UPLC-MS in berries. The BTH-treated vines showed significantly higher level of *E*-piceid compared to the controls at middle veraison and harvest stages of both vintages (Appendix A). Specifically, the BTH high dose provided grapes with higher levels of *E*-piceid than the control and BTH low dose at middle veraison stage of both 2019 and 2020 vintages. The highest *E*-piceid content among the treatments was recorded at harvest stages of 2019 and 2020 upon the application of high and low doses of BTH, respectively. Similar results were obtained upon the exogenous application of benzothiadiazole in the red-colored variety, Merlot [47], and chitosan and abscisic acid in the white-colored variety, Savvatiano [32].

### 2.8. Physicochemical Parameters of Mouhtaro Experimental Wines

Benzothiadazole treatment had no effect on the alcoholic degree in 2019 wine samples. Contrarily, the alcoholic title was significantly lower in the wines produced from treated vines compared to the controls in 2020 vintage (Table 6), which is similar to Savvatiano wines (Table 2). These results are in agreement with Vitalini et al. [29] in which the BTH-treated vines of the red-colored variety, Groppello Gentile, produced wines with a lower alcoholic title. These results paralleled the physiochemical results of the Mouhtaro grapes. Regarding the TA, BTH treatments reduced the total acidity of the wines with respect to the control wines and increased the pH levels. As for the volatile acidity, BTH treatments increased the volatile acidity; however, the values were well below 0.6 g/L (Table 6). 

### 2.9. Color and Phenolic Characteristics of the Mouhtaro Experimental Wines

The enological parameters such as Total Phenolic Index, wine color intensity, total anthocyanins, Total Polyphenolic Concentration value, and the total tannins measured by the Methyl Cellulose Precipitable (MCP) assay revealed differences among control wines and wines made from BTH-treated vines in the two seasons (Table 7). The BTH-treated vines produced wines with significantly higher color and phenolic profile when compared to the control wines in both vintages. However, the concentration of the phenolics recorded in wines was decreased in 2020 compared to 2019 vintage, probably due to the higher precipitation. Therefore, the effect of the BTH treatment on the phenolics and the enological parameters were milder than those detected for the 2019 wine samples. Similar differences due to climatological conditions were also observed [52] in the Tempranillo variety in the same vintages. Furthermore, Ruiz- Garcia et al. [35] observed an increase in athocyanin and phenolic concentration in grapes and further to the elaborated wines upon the biostimulant application. 

### 2.10. Anthocyanins Content (mg/L) in Mouhtaro Experimental Wines

The color of grapes and red wines is due to the presence of anthocyanins. Anthocyanins biosynthesis occurs in the skin, and it is a critical factor of the diversity of the varieties [53]. The wine anthocyanin content obtained by HPLC was different to that analyzed in berries, probably due to the differential extraction ratio of anthocyanins, degradation, and polymerization reactions [31,54].

In both vintages, the monomeric anthocyanins in the BTH-treated wines showed significantly increased levels than the control wines. More specifically, the highest content among the treatments was recorded in BTH low dose (Table 8). Moreover, the concentration of anthocyanins in wines was decreased in response to the higher precipitations observed during the year 2020 (Table 1). In addition to that, malvidin-3-*O*-glucoside significantly scored the highest content in all BTH treatments in the two consecutive seasons. The enhanced content of malvidin*-3-O*-glucoside was also reported upon the exogenous application of BTH and Methyl Jasmonate (MeJ) in Syrah and Tempranillo varieties [30]. The total anthocyanin content obtained in our study is in agreement with the increase in the phenolic capacity of both grapes and wines. The results also confirm previous studies using the foliar application of ΒΤH in red varieties Monastrell, Merlot, and Cabernet Sauvignon [27], and in Monastrell wines [26,35]. The total content of individual monomeric anthocyanins (as quantified by HPLC) in wines also correlates with the wine absorbance (color intensity and total anthocyanins).

### 2.11. Gene Expression

To investigate the molecular nature of the metabolic shifts observed in grape berries upon BTH treatments, the expression of genes encoding key intermediates of core metabolic pathways was studied. Recent results have highlighted that benzothiadiazole caused alterations in grape berry transcriptome [38,55,56,57]. In the present study, the effect of benzothiadiazole application on gene expression was examined by the targeted RT-qPCR analysis of berry samples collected at three different stages (veraison, middle veraison, and harvest) during the 2019 and 2020 vintages. 

We initially investigated the expression profile of phenylalanine ammonia lyase (*VviPAL*) and cinnamate 4-hydrolase (*VviC4H*)—genes that encode the first two enzymes of the phenylpropanoid pathway [58]. The expression level of the *VviPAL* gene in control vines was higher in the red-colored Mouhtaro than the white-colored Savvatiano (Figure 3a). Benzothiadiazole application had no statistically significant effect on the *VviPAL* expression, although the trend was to increase the expression level, especially the second vintage in Mouhtaro (Figure 3a). The *VviC4H* transcript level in control vines was increased after veraison in both vintages in Mouhtaro but remained constant during maturation in Savvatiano (Figure 3b). The application of BTH resulted in an increased *VviC4H* expression level at all phonological stages in Mouhtaro and at the middle veraison stage in Savvatiano (Figure 3c,d). 

The expression of UDP-glucose-flavonoid 3-*O*-glycosyltransferase gene (*VviUFGT*), encoding for the critical step in anthocyanin biosynthesis [59], was found to be higher in the red-colored Mouhtaro compared to the white-colored Savvatiano (Figure 3c). Similarly to *VviC4H*, the *VviUFGT* expression was increased by BTH application at all phenological stages in Mouhtaro, while the effect was negligible and dependent on the vintage in the Savvatiano variety (Figure 3b,c). In the final step of anthocyanin synthesis, all the genes of the flavonoid pathway were present both in white and red grape berries, including a UDP glucose-flavonoid 3-*O*-glucosyl transferase (UFGT) [60,61]. Moreover, Walker et al. [61] found that two very similar regulatory genes, VvMYBA1 and VvMYBA2, which could activate anthocyanin biosynthesis, were not transcribed in white skin berries [62]. Nevertheless, the existence of several other MYB-type transcription factors that can modulate flavonoid biosynthesis [63] and the identification of VviUFGT in transcriptomic studies in white-colored cultivars [64] imply more complicated regulatory mechanisms.

The stilbene synthase gene (*VviSTS*), initiating the stilbenes biosynthesis [65], also exhibited a cultivar-dependent expression profile, with obviously higher transcript accumulation in Savvatiano (Figure 3d). Benzothiadiazole treatment resulted in the up-regulation of the *VviSTS* expression at the middle veraison stage of both vintages in Savvatiano. On the other hand, an increased expression level in treated vines of Mouhtaro was observed only in the second vintage (Figure 3d).

We further examined the expression patterns of genes involved in the biosynthesis of flavonols and flavan-3-ols. The expression of flavonol synthase gene (*VviFLS*) that encodes for the enzyme catalyzing the flavonol biosynthesis [66] was significantly higher in the white-colored Savvatiano. Benzothiadiazole applications caused an alteration in the *VviFLS* expression level, mostly dependent on the vintage (Appendix A). A similar vintage-dependent response to BTH—and a lower expression level in the control vines of both cultivars—was observed for leucoanthocyanin reductase 1 gene (*VviLAR1*) and anthocyanidin reductase (*VviANR*), both involved in flavan-3-ols biosynthesis [67] (Appendix A).

Altogether, the results observed by gene expression analysis suggest that the application of benzothiadiazole had a positive effect on the *VviC4H* expression in both cultivars, on the *VviUFGT* expression in Mouhtaro and on the *VviSTS* expression in Savvatiano. Minor alterations were also observed in the expression patterns of the other genes examined, but they were mostly vintage-dependent.

## 3. Discussion

The application of biostimulants on grapevine is a promising technique which improves the quality of grapes and the produced wine, specifically during recent years in which viticulture has faced the challenge of climate change. The application of a biostimulant onto grapevine during the developmental stage is crucial in order to optimize the result. Herein, we investigated the influence of benzothiadiazole on the development of key phenolic compounds and the expression level of specific key genes from the phenylpropanoid pathway during the ripening period, using targeted UPLC- MS and RT-qPCR analysis.

The higher mean temperature during the post-veraison stage of 2020, combined with higher rainfall than in 2019 (Table 1), produced grape berries with higher levels of the weight/berry ratio. Consequently, the content of the phenolic compounds of the berries in the 2020 vintage were lower than those in 2019, possibly by inhibition of their biosynthesis. In the two consecutive vintages, BTH did not substantially affect the majority of berry quality phenolic compounds.

BTH application increased the expression level of the *VviC4H*, *VviUFGT* genes in Mouhtaro and the *VviC4H*, *VviSTS* genes in Savvatiano in both years. The changes in the expression of the remaining genes were found to be dependent on the vintage. It should be mentioned that the gene expression levels reflect the content of the phenolic compounds. Similar alterations in the expression of phenylpropanoid pathway genes caused by biostimulant application have been also reported in recent transcriptome studies [68,69,70].

Regarding the outcome from the gene expression studies, it was revealed that patterns were related to the plant’s developmental stage regardless of the environmental/climatic conditions. Characteristic examples of developmental regulation can consider the expression pattern of *VviSTS* (stilbenoids) [64], which was found to gradually increase during ripening in both cultivars, while the expression of *VviLAR1,* which is a key regulator of the flavan-3-ol [71], decreased during the ripening period in both cultivars. In contrast, the vintage effect was evident in the expression patterns of the remaining genes studied. The trend was similar to the content levels of the additional phenolic compounds during their development in both studied years. Several studies conducted in recent years have documented the application of bioelicitors and recorded grapes with an enhanced content of athocyanins and stilbenes [12,51,71,72]. Therefore, several elicitors that are applied exogenously have the capacity to increase the content of the phenolic compounds due to the activation of the enzymes involved in their synthesis, mainly affecting enzymes specifically related to anthocyanins and stilbenes [73,74].

The wines produced in 2019 and 2020 from grapes treated with BTH presented differences in the content of phenolic compounds with respect to the control wines. However, it should be mentioned that wines recorded less phenolic compounds in 2020 vintage compared to 2019, which was due to the higher rainfall. These results are vividly shown in the white-colored variety. Considering the effect of treatments, the positive effects of BTH that were observed in the grapes were also reflected in the wines. The wines produced from BTH-treated grapes as well, with respect to weather conditions, may have an even greater impact on wine’s chromatic and phenolic properties.

According to our results, in the red-colored variety, the wines produced from treated grapes had higher color intensity and total phenolic content than the wines produced from the control grapes, indicating that the BTH application may be useful in producing wines with an intense and stable red color. BTH application to Monastrell variety showed a similar pattern [26,27]. A similar phenolic profile was also observed in the white-colored variety in 2019 vintage, i.e., BTH-treated wines recorded higher phenolic content (TPI, Folin–Ciocalteau’s assay, 420 nm absorbance and k factor) than the control wines.

Conclusively, the exogenous application of biostimulants such as benzothiadiazole can be used in order to induce the biosynthesis of secondary metabolites of oenological interest, with the aim of improving the quality characteristics of grapes and further to produce wine styles that the winemaker will choose. The timing of the biostimulants application has a significant impact on which compounds are dominant until harvest.

## 4. Materials and Methods

### 4.1. Experimental Design and Biostimulant Application

The experiment was conducted during two growing seasons (2019 and 2020) in a commercial vineyard in Muses Valley for the Mouhtaro and Savvatiano varieties at an elevation of 450 m in Central Greece. The Savvatiano vines were more than 50 years old and pruned as bush vines. Mouhtaro vines were 12 years old. They were trained on double cordon and the pruning system was 3 spurs in each cordon.

The vineyards were managed according to standard agronomical practices of the region, without irrigation. The number and timing of viticultural practices (i.e., plant protective applications) were similar for all treatments in each variety. The experiments were conducted in a randomized block design according to [32]. Vines of each variety were sprayed with an aqueous solution of (i) 0.3 mM (low application dose) and (ii) 0.6 mM (high application dose) benzothiadiazole (benzo-(1,2,3)-thiadiazole-7-carbothioic acid S-methyl ester, BTH, trade name Bion^®^, Syngenta^®^, Basel, CH, USA) or water (control). Wetting agent Tween 80 (Sigma–Aldrich) (St. Louis, MO, USA) was used. BTH application was carried out in the whole vine canopy at veraison stage and then 7 and 14 days later (Appendix A). Three different replicates were applied in each treatment.

### 4.2. Physicochemical Results of the Must and Wine

Grape samples were obtained during the two vintages from the commercial monovarietal vineyard Lachos (Mouhtaro single vineyard 38°19′30.9″ (N) and 23°05′37.7″ (E)) and Papanicolas (Savvatiano single vineyard, 38°19′30″ (N), 23°05′37″ (E)) at the three phenological stages (veraison, middle veraison, and harvest) (Appendix A). Samples of 50 berries were collected randomly from each plot per sampling date and the fresh weight of the berries was determined. Grape berry juice was analyzed for total soluble solids (TSS in °Brix) by refractometry, titratable acidity (g/L of tartaric acid), and pH according to the OIV [75].

### 4.3. Grape Metabolic Profile by UPLC-MS

The frozen grape berries (−80 °C) were ground to powder with liquid nitrogen after the elimination of the seeds and used for metabolic profiling, based on the adapted methods from previous studies [76,77]. A total of 50 mg of ground-to-powder berry dry weight was extracted using 1 mL of 80% (*v*/*v*) methanol. After 30 min of sonication, the samples were macerated overnight at 4 °C in the dark and centrifuged at 18,000× *g* for 10 min. The supernatant was diluted five-fold in 80% (*v*/*v*) methanol and stored at −20 °C prior to further analyses. The UPLC–MS was performed using an ACQUITY™ Ultra Performance Liquid Chromatography system coupled to a photo diode array detector (PDA) and a Xevo TQD mass spectrometer (Waters, Milford, MA, USA), equipped with an electrospray ionization (ESI) source controlled by Masslynx 4.1 software (Waters, Milford, MA, USA). The analyte separation was achieved by using a Waters Acquity HSS T3 (C18) column (150 × 2.1 mm, 1.8 μm), with a flow rate of 0.4 mL min^−1^ at 55 °C. The injection volume was 5 μL. The mobile phase consisted of solvent A (0.1% formic acid in water) and solvent B (0.1% formic acid in acetonitrile). The chromatographic separation was achieved using an 18 min linear gradient from 5 to 50% solvent B. The MS detection was performed in both the positive and negative modes. The capillary voltage was 3000 V and the sample cone voltages were 30 and 50 V. The cone and desolvation gas flow rates were 60 and 800 Lh−1. The identification of the analytes was based on retention times, *m*/*z* values and UV spectra, and by comparison with commercial standards, own purified compounds, or data from the literature when no authentic standards were available, as reported in Milliordos et al. [32]. Data collection was carried in selected ion monitoring (SIM) mode for the following compounds in positive mode: L-proline (*m*/*z* 116); L-isoleucine and L-leucine (*m*/*z* 132), L-phenylalanine (*m*/*z* 166); L-tyrosine (*m*/*z* 182); L-tryptophan (*m*/*z* 205); cyanidin-3-*O*-galactoside (*m*/*z* 449); delphinidin 3-*O*-glucoside (*m*/*z* 465); petunidin-3-*O*-glucoside (*m*/*z* 479); cyanidin-3-*O*-(6-*O*-acetyl)-glucoside (*m*/*z* 491); malvidin-3-*O*-glucoside (*m*/*z* 493); peonidin-3-*O*-(6-acetylglucoside)) (*m*/*z* 505); petunidin-3-*O*-(6-*O*-acetyl)-glucoside (*m*/*z* 521); malvidin-3-*O*-(6-*O*-acetyl)-glucoside (*m*/*z* 535); peonidin-3-*O*-(6-*p*-coumaroyl-glucoside)) (*m*/*z* 609); petunidin-3-*O*-(6-*p*-coumaroyl)-glucoside (*m*/*z* 625); malvidin-3-*O*-(6-*p*-coumaroyl)-glucoside (*m*/*z* 639); malvidin-3,5-*O*-diglucoside (*m*/*z* 655); peonidin-3-*O*-(6-*p*-coumaroyl-glucoside) (*m*/*z* 609). Additionally, the following compounds were targeted in positive mode; gallic acid (*m*/*z* 169); catechin and epicatechin (*m*/*z* 189); coutaric acid (*m*/*z* 295); caftaric acid (*m*/*z* 311); fertaric acid (*m*/*z* 325); *E*-piceid (*m*/*z* 389); catechin-gallate (*m*/*z* 441); kaempferol-3-*O*-glucoside (*m*/*z* 447); quercetin-3-*O*-glucoside (*m*/*z* 463); quercetin-*O*-glucuronide (*m*/*z* 477); myricetin-glucoside (*m*/*z* 479); procyanidin B1-B4 (*m*/*z* 577); procyanidin-gallate (*m*/*z* 729); *E*-resveratrol (*m*/*z* 227); *E*-piceatannol (*m*/*z* 243); *E*-ε-viniferin (*m*/*z* 453); kaempferol-3-*O*-rutinoside (*m*/*z* 593). Absolute quantifications were conducted using pure standards and using a five-point calibration curve (0–20 ppm). Total amino acid concentrations represented the sum of L-isoleucine, L-leucine, L-phenylalanine, L-tyrosine, and L-tryptophan concentrations. Total anthocyanin diOH concentrations represented the sum of cyanidin-3-*O*-galactoside and peonidin-3-*O*-(6-*p*-coumaroyl-glucoside). Total stilbenoid concentration represented the sum of *E*-piceid, *E*-resveratrol and *E*-piceatannol concentrations. Total flavan-3-ol concentrations represented the sum of catechin, epicatechin, catechin gallate, procyanidin B1-4 and procyanidin gallate.

The extraction and UPLC–MS analyses were performed in triplicates.

### 4.4. RNA Extraction and Analysis of Gene Expression

Grape berries without seeds were ground to powder; liquid nitrogen and RNA was extracted Reid et al. [78]. Briefly, approximately 1 g of ground tissue was extracted with buffer containing 300 mM Tris-HCl (pH 8.0), 25 mM EDTA, 2 M NaCl, 2% (*w*/*v*) CTAB, 2% (*w*/*v*) PVPP, and 0.05% (*w*/*v*) spermine at 65 °C for 15 min, mixed thoroughly with an equal volume of chloroform: isoamyl alcohol (24:1), and centrifuged. The step was repeated for the aquatic phase; the RNA was precipitated with a 0.6 volume of isopropanol and a 0.1 volume of sodium acetate at −20 °C overnight, centrifuged, and finally dissolved in 100 μL ddH_2_O. The RNA samples were treated with DNAse I (Takara Bio, Shiga, Japan) and further purified using phenol: chloroform: isoamyl alcohol (25:24:1) followed by ethanol precipitation. The RNA quantity and quality were determined using a NanoDrop ND-1000 Spectrophotometer (Thermo Fisher Scientific Inc., Wilmington DE, USA) and verified by 0.8% agarose gel electrophoresis. Reverse transcription was performed with 2 μg RNA using SMART MMLV-Reverse Transcriptase (Takara Bio, Shiga, Japan) and oligo (dT) primer (Eurofins Genomics, Ebersberg, Germany). The synthesized cDNA was five-fold diluted and PCR conditions were optimized for primers corresponding to selected genes from the phenylpropanoid pathway [32]. The samples were further diluted and quantitative PCR reactions were performed in the PikoReal Real-Time PCR System (Thermo Fisher Scientific, Vantaa, Finland) using KAPA SYBR FAST qPCR Master Mix (KAPA Biosystems, Cape Town, South Africa) and by applying the following cycler conditions: 2 min at 50 °C, 2 min at 95 °C, followed by 40 cycles of 15 s at 95 °C, 30 s at 62 °C, 30 s at 72 °C. All quantitative PCR reactions were performed as triplicates and melting curve analysis was performed at the end of each reaction to confirm primer specificity. The quantification of gene expression was performed according to the 2-ΔCt method and elongation factor 1a (*VviEF1a*) was used as the reference gene for data normalization.

### 4.5. Vinification Process

Grapes from the two varieties were harvested in their optimum technological maturity. For each replicate, 25 kg of grapes were handpicked and transferred to the experimental winery of the Laboratory of Enology and Alcoholic Drinks at the Agricultural University of Athens, and stored overnight at 4 °C. The vinification process started the day after harvest early in the morning. Standard white and red vinification protocols were used as described by Miliordos et al. [79,80].

### 4.6. Wine Phenolics

#### 4.6.1. Wine Color, Color Density

According to Sudraud [81], color intensity was determined through molecular absorbance measured at 420, 520, and 620 nm. The color intensity gives, thus, an estimation of the total color of a sample [82]. All analyses were performed in triplicate.

#### 4.6.2. Total Anthocyanins

Total anthocyanins were determined by a spectrophotometric method based on SO_2_ bleaching [83], named the Bisulfite Bleaching method. All analyses were performed in triplicate.

#### 4.6.3. Total Phenolic Index, Follin–Ciocalteau, and Browning Test

The Total Polyphenol Index (TPI) was determined by measuring the 280 nm absorbance of a 1:100 dilution of red wine and 1:25 of white wine with a spectrophotometer, using a 10 mm quartz cuvette and multiplying the absorbance value by 100 [84].

The total polyphenol concentration was determined with the Folin–Ciocalteau assay using the microscale protocol [85]. The results were expressed as mg/L of gallic acid equivalents.

The model used to assess browning development was a modification of the model described by Singleton and Kramling [86]. Wine lots of 30 mL were filtered and placed in a 30 mL, screw-cap glass vial (7.5 cm length, 2.1 cm internal diameter). Samples were subjected to heating at a constant temperature of 55.0 ± 0.2 °C in a heating chamber. Aliquots were withdrawn at 24 h intervals over a period of 13 days, and browning (A_420_) was measured. The samples were then immediately returned to the vials to maintain the initial headspace volume.

#### 4.6.4. Anthocyanins by HPLC

Determination of the wine monomeric anthocyanins was carried out according to Miliordos et al. [81]. In detail, reversed-phase HPLC analyses of anthocyanins were carried out by a direct injection of 10 μL of wine into a Waters 2695 Alliance liquid chromatograph system coupled with a Waters 2996 PDA detector (Milford, MA, USA) and by using a SVEA C18 Plus 4.6 × 250 mm, 5 μm column (Nanologica, Södertälje, Sweden). The mobile phases used were 10% aqueous formic acid (solvent A) and methanol (solvent B). Chromatograms were recorded at 520 nm, and anthocyanin standard curves were made using malvidin-3-*O*-glucoside chloride. Identification was based on comparing the retention times of the peaks detected with those of original compounds, and on a UV–Vis spectrum. The anthocyanidin-3-*O*-monoglucosides delphinidin, peonidin, petunidin, and malvidin and the acetylated and *p*–coumarylated of malvidin were expressed as mg/L of malvidin-3-*O*-glucoside.

#### 4.6.5. Tannin Determination with Methyl Cellulose Precipitation (MCP) Assay

The MCP analysis assay was performed according to Sarneckis et al. [87]. MCP was carried out in the red wines, in both vintages (2019–2020). All analyses were performed in triplicate.

#### 4.6.6. Climate Conditions

Meteorological data were obtained during the experiment from the National Observatory of Athens Automatic Network [88].

## 5. Conclusions

The results of this study indicate that the exogenous application of BTH during veraison could induce the biosynthesis of significant phenolic compounds. Targeted metabolomic analysis revealed that anthocyanins and stilbenes were increased in both varieties, which concurs with the corresponding gene expression profile. Particularly, the increase in stilbene compounds concentration is of great interest considering their significant antioxidant activity. According to the results obtained in the two-year trial, the effect of the biostimulant application can be already observed immediately after the second sampling. Furthermore, our results showed that benzothiadiazole affected the phenolic compounds of the produced wines as well. In particular, BTH increased the phenolic and chromatic profile of the produced experimental wines in both vintages. The results indicate that the application of benzothiadiazole deserves attention beyond its efficacy in crop protection, regarding its effect on the overall quality of the produced wine. The exogenous application of benzothiadiazole could be utilized to induce the biosynthesis of secondary metabolites of oenological interest, and therefore, to improve the quality characteristics of grapes and the corresponding wine styles that the winemaker will choose to produce. The timing of application is likely to be an important factor for these compounds to prevail until harvest.

## Figures and Tables

**Figure 1 plants-12-01179-f001:**
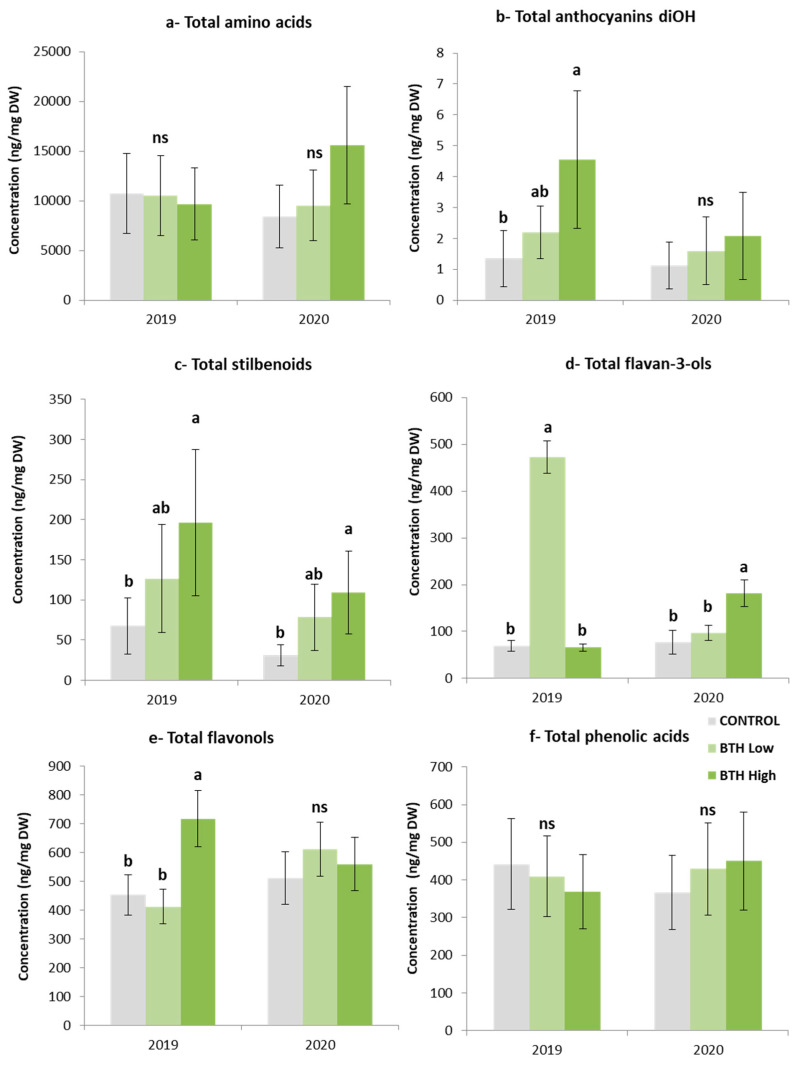
Total concentrations of amino acids (**a**), anthocyanins diOH (**b**), stilbenoids (**c**), flavan-3-ols (**d**), flavonols (**e**) and phenolic acids (**f**) in Savvatiano berries at harvest stage (3rd sampling) in 2019 and 2020 treated with benzothiadiazole. Control (grey), low concentration of benzothiadiazole (light green), and high concentration of benzothiadiazole (dark green). Error bars represent the standard deviations. Different letters indicate significant differences No significant difference (ns) was found between values with the same letters (one-way ANOVA, *p*-value > 0.05).

**Figure 2 plants-12-01179-f002:**
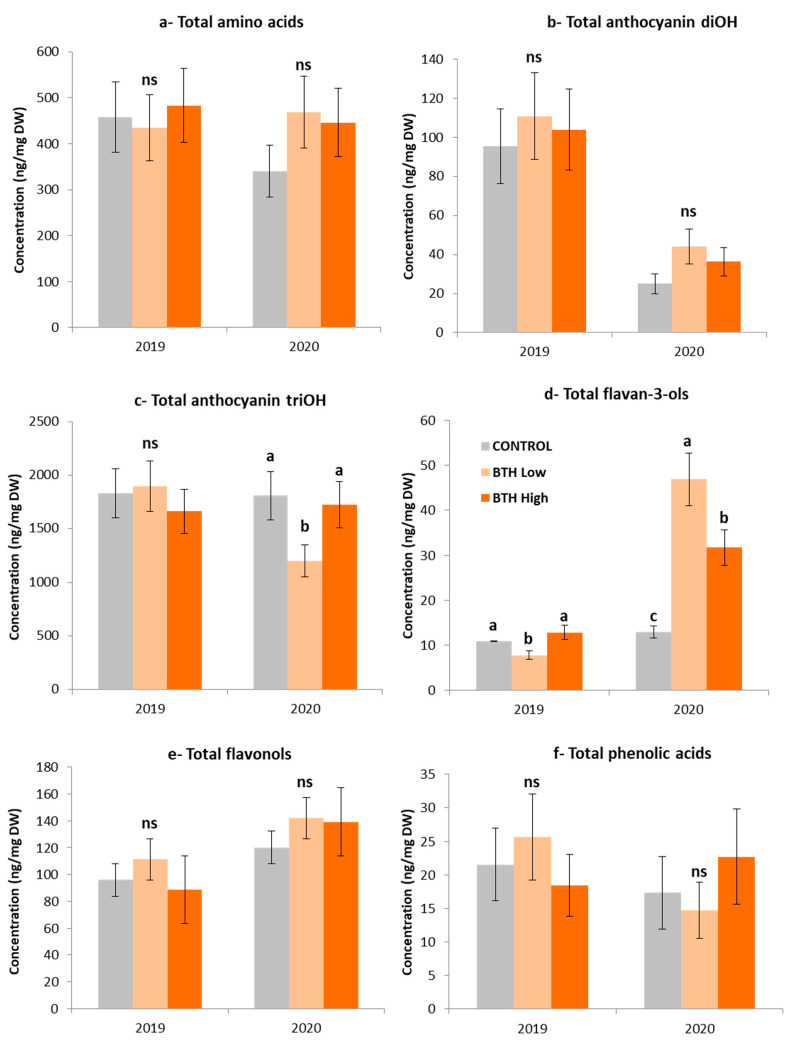
Total concentrations of amino acids (**a**), anthocyanins diOH (**b**), stilbenoids (**c**), flavan-3-ols (**d**), flavonols (**e**), and phenolic acids (**f**) in Mouhtaro berries at Harvest (3rd sampling) stage in 2019 and 2020 treated with benzothiadiazole: control (grey), low concentration of benzothiadiazole (pale orange), and high concentration of benzothiadiazole (dark orange). Error bars represent the standard deviations. Different letters indicate significant differences. No significant difference (ns) was found between values with the same letters (one-way ANOVA, *p*-value > 0.05).

**Figure 3 plants-12-01179-f003:**
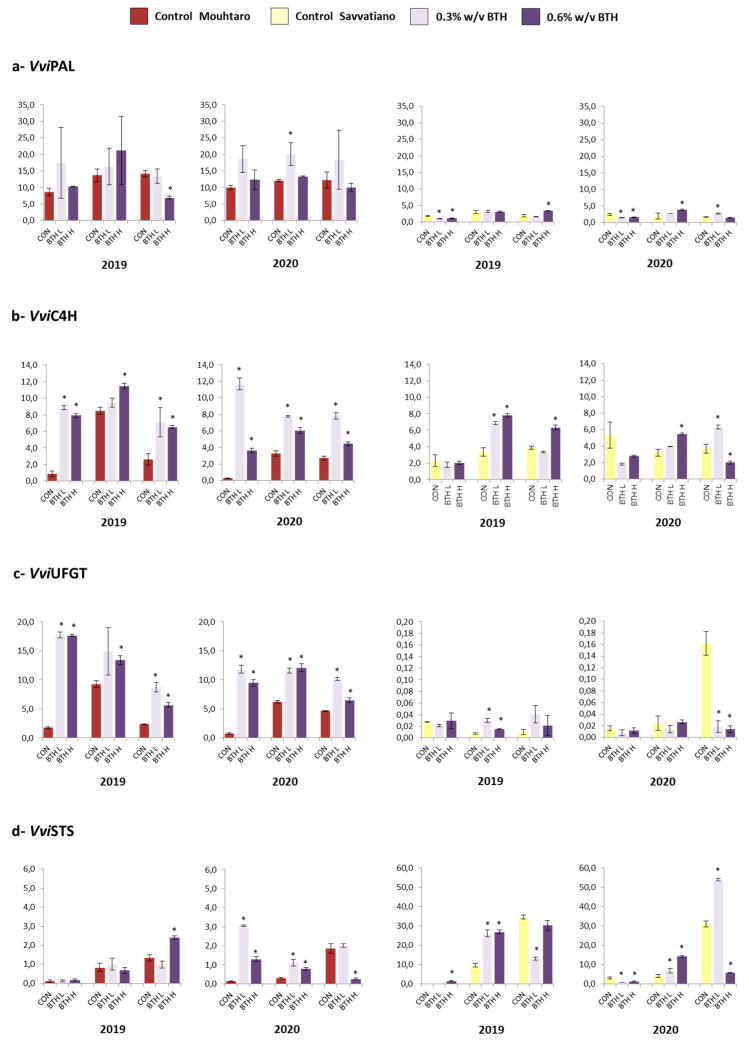
The expression level of genes involved in the phenylpropanoid pathway: *VviPAL* (**a**), *VviC4H* (**b**), *VviUFGT* (**c**), and *VviSTS* (**d**) in Mouhtaro and Savvatiano during two growing seasons (2019 and 2020). Vertical bars represent the standard deviation and asterisks indicate the statistically significant differences (Student’s *t*-test. *p*–value < 0.05). The three sampling points (veraison, middle veraison, and harvest) are indicated under each graph.

**Table 1 plants-12-01179-t001:** Meteorological data of the Muses Valley area during 2019 and 2020 vintages.

Meteorological Data					
	Savvatiano			Mouhtaro	
		2019	2020	2019	2020
Mean Temperature (°C)	Pre-veraison ^1^	22.8	22.2	19.62	19.42
	Post-veraison ^2^	22.6	23.8	24.7	26
	Growing season ^3^	22.8	22.5	21.11	22.32
Rainfall (mm)	Pre-veraison	363.8	266.5	146.2	207.4
	Post-veraison	16.8	5.2	0.0	84.0
	Growing season	112.0	299.2	146.2	291.4
	Annual ^4^	492.6	570.9	505.4	587.5

^1^ Pre-veraison, period from budbreak to veraison; ^2^ Post-veraison, period from veraison to harvest; ^3^ Growing season, period from budbreak to harvest; ^4^ Annual, period from harvest to harvest.

**Table 2 plants-12-01179-t002:** Physicochemical characteristics of Savvatiano grapes at three phenological stages.

**2019**					
**Phenolic ** **Stage**	**Treatment**	**Berry Volume (mg/berry)**	**Total Soluble Solids (^o^Brix)**	**Total Acidity (Tartaric Acid g/L)**	**pH**
V	Control	2.21 ± 0.29 ab	14.26 ± 0.75 b	5.42 ± 0.24 b	2.89 ± 0.12 a
	BTH Low	2.23 ± 0.16 a	14.95 ± 0.30 ab	5.70 ± 0.32 ab	2.77 ± 0.02 a
	BTH High	1.83 ± 0.05 b	15.3 ± 0.17 a	6.15 ± 0.22 a	2.87 ± 0.15 a
MV	Control	2.70 ± 0.16 a	17.13 ± 0.55 b	4.75 ± 0.11 a	3.14 ± 0.07 b
	BTH Low	2.39 ± 0.12 b	18.13 ± 0.28 a	4.45 ± 0.35 a	3.32 ± 0.04 a
	BTH High	2.07 ± 0.06 c	18.66 ± 0.11 a	4.72 ± 0.19 a	3.35 ± 0.01 a
H	Control	2.30 ± 0.06 b	19.53 ± 0.11 a	4.40 ± 0.67 a	3.35 ± 0.09 a
	BTH Low	2.45 ± 0.07 a	19.73 ± 0.49 a	4.35 ± 0.30 a	3.37 ± 0.03 a
	BTH High	2.12 ± 0.03 c	19.96 ± 0.35 a	3.90 ± 0.25 a	3.39 ± 0.07 a
**2020**					
**Phenolic** **Stage**	**Treatment**	**Berry volume (mg/berry)**	**Total Soluble Solids (^o^Brix)**	**Total Acidity (Tartaric Acid g/L)**	**pH**
V	Control	2.64 ± 0.05 a	15.66± 0.73 a	6.2 ± 0.17 a	3.25 ± 0.03 a
	BTH Low	2.63 ± 0.06 a	14.83 ± 0.64 a	6.4 ± 0.44 a	3.19 ± 0.09 a
	BTH High	2.56 ± 0.10 a	15.00 ± 0.17 a	6.5 ± 0.43 a	3.20 ± 0.04 a
MV	Control	2.85 ± 0.11 a	17.76 ± 0.28 a	5.3 ± 0.34 a	3.32 ± 0.09 a
	BTH Low	2.58 ± 0.07 b	16.53 ± 0.89 a	5.45 ± 0.22 a	3.22 ± 0.02 a
	BTH High	2.59 ± 0.08 b	16.50 ± 0.70 a	5.6 ± 0.31 a	3.24 ± 0.04 a
H	Control	2.56 ± 0.23 a	20.13 ± 0.40 a	4.49 ± 0.26 a	3.36 ± 0.03 a
	BTH Low	2.82 ± 0.19 a	19.53 ± 0.25 ab	4.6 ± 0.17 a	3.37 ± 0.07 a
	BTH High	2.55 ± 0.14 a	19.10 ± 0.25 b	4.63 ± 0.20 a	3.39 ± 0.13 a

Data represent mean ± std. deviation (*n* = 3). For each vintage and compound, the mean values followed by different letters in the same column are significantly different according to the *t*-test at 5% probability. Abbreviations: V—Veraison; MV—Mid-Veraison, and H—Harvest).

**Table 3 plants-12-01179-t003:** Conventional wine analysis parameters of experimental Savvatiano wines in response to two different treatments (ΒΤH low and high) and a control.

Savvatiano					
Vintage	Treatment	Ethanol (*v*/*v* %)	Total Acidity (Tart. Acid g/L)	Volatile Acidity (Ac. Ac. g/L)	pH
2019	Control	10.8 ± 0.2 a	6.0 ± 0.3 a	0.2 ± 0.04 a	3.10 ± 0.04 a
	BTH Low	10.8 ± 0.3 a	6.0 ± 0.1 a	0.2 ± 0.03 a	3.20 ± 0.12 a
	BTH High	11.5 ± 0.5 a	6.2 ± 0.3 a	0.2 ± 0.01 a	3.17 ± 0.07 a
2020	Control	11.8 ± 0.2 a	4.1 ± 0.1 a	0.10 ± 0.01 a	3.47 ± 0.02 a
	BTH Low	10.8 ± 0.1 b	4.3 ± 0.1 a	0.10 ± 0.02 a	3.35 ± 0.01 a
	BTH High	10.7 ± 0.2 b	4.6 ± 0.5 a	0.10 a ± 0.06 a	3.33 ± 0.11 a

Data represent mean ± std. deviation (*n* = 3). Data represent mean ± std. deviation (*n* = 3). For each vintage and analysis, the mean values followed by different letters in the same column are significantly different according to the *t*-test at 5% probability.

**Table 4 plants-12-01179-t004:** Color and phenolic parameters of the three experimental Savvatiano wines in response to two different treatments.

Vintage	Treatment	TPI	420 nm	Total Polyphenol Concentration (Gal. Ac. mg/L)	k Factor
Savvatiano 2019	Control	5.37 ± 0.38 b	0.047 ± 0.003 b	23.2 ± 1.0 b	0.0031 ± 0.0005 a
	BTH Low	5.72 ± 0.46 ab	0.0513 ± 0.007 ab	25.0 ± 2.0 b	0.0035 ± 0.0002 a
	BTH High	6.42 ± 0.47 a	0.058 ± 0.002 a	31.1 ± 1.5 a	0.0036 ± 0.0001 a
Savvatiano 2020	Control	5.23 ± 0.06 a	0.05 ± 0.005 a	25.2 ± 1.6 a	0.060 ± 0.0004 b
	BTH Low	4.46 ± 0.31 b	0.04 ± 0.005 b	20.6 ± 1.9 b	0.0641 ± 0.005 a
	BTH High	4.45 ± 0.37 b	0.04 ± 0.001 b	21.5 ± 1.4 b	0.067 ± 0.002 a

Data represent mean ± std. deviation (*n* = 3). For each vintage and analysis, the mean values followed by different letters in the same column are significantly different according to the *t*-test at 5% probability.

**Table 5 plants-12-01179-t005:** Physiochemical characteristics of Mouhtaro grapes at three phenological stages.

**2019**					
**Phenologic Stage**	**Treatment**	**Berry Volume (mg/berry)**	**Total Soluble Solids (^o^Brix)**	**Total Acidity (Tartaric Acid g/L)**	**pH**
V	Control	1.51 ± 0.15 a	14.1 ± 0.8 b	12.3 ± 0.4 a	2.86 ± 0.03 a
	BTH Low	1.39 ± 0.10 ab	15.9 ± 0.1 a	13.0 ± 0.6 a	2.88 ± 0.13 a
	BTH High	1.22 ± 0.02 b	16.2 ± 1.1 a	12.1 ± 0.2 a	2.87 ± 0.15 a
MV	Control	2.02 ± 0.10 a	18.9 ± 1.6 a	9.5 ± 1.1 a	3.50 ± 0.02 a
	BTH Low	1.77 ± 0.05 b	19.1 ± 0.3 a	9.2 ± 0.2 a	3.41 ± 0.16 a
	BTH High	1.59 ± 0.02 b	18.0 ± 0.8 a	9.4 ± 0.6 a	3.50 ± 0.05 a
H	Control	2.11 ± 0.01 a	23.7 ± 0.4 a	8.1 ± 0.2 a	3.31 ± 0.01 a
	BTH Low	2.10 ± 0.07 ab	23.4 ± 0.3 a	8.1 ± 0.1 a	3.69 ± 0.07 b
	BTH High	2.01 ± 0.02 b	23.2 ± 0.5 a	7.4 ± 0.2 b	3.76 ± 0.02 b
**2020**					
**Phenolic ** **Stage**	**Treatment**	**Berry volume (mg/berry)**	**Total Soluble Solids (^o^Brix)**	**Total Acidity (Tartaric Acid g/L)**	**pH**
V	Control	1.74 ± 0.12 a	14.7 ± 0.2 a	17.9 ± 0.1 a	2.97 ± 0.06 a
	BTH Low	1.88 ± 0.10 a	15.0 ± 0.8 a	16.8 ± 0.4 b	3.05 ± 0.05 a
	BTH High	1.90 ± 0.11 a	14.3 ± 0.1 a	17.4 ± 0.2 ab	3.08 ± 0.02 a
MV	Control	2.38 ± 0.06 a	17.7 ± 0.3 a	10.3 ± 0.4 a	3.21 ± 0.09 a
	BTH Low	2.16 ± 0.19 ab	18.3 ± 0.5 a	10.3 ± 0.4 a	3.23 ± 0.03 a
	BTH High	2.06 ± 0.10 b	17.6 ± 0.3 a	10.8 ± 0.2 a	3.16 ± 0.03 a
H	Control	2.05 ± 0.09 a	24.6 ± 0.5 a	7.6 ± 0.3 a	3.37 ± 0.12 b
	BTH Low	1.90 ± 0.23 a	23.6 ± 0.3 b	8.7 ± 0.9 a	3.51 ± 0.09 ab
	BTH High	1.96 ± 0.80 a	23.6 ± 0.2 b	8.6 ± 0.2 a	3.62 ± 0.08 a

Data represent mean ± std. deviation (*n* = 3). For each vintage and analysis, the mean values followed by different letters in the same column are significantly different according to the *t*-test at 5% probability. Abbreviations: V—Veraison, MV—Mid-Veraison, and H—Harvest).

**Table 6 plants-12-01179-t006:** Conventional wine analysis parameters of experimental Mouhtaro wines in response to two different BTH treatments (low and high).

Vintage	Treatment	Ethanol (*v*/*v* %)	Total Acidity (Tart. Acid g/L)	pH	Volatile Acidity (Ac. Ac. g/L)
Mouhtaro 2019	Control	13.7 ± 0.4 a	6.2 ± 0.3 a	3.43 ± 0.04 a	0.43 ± 0.07 a
	BTH Low	13.4 ± 0.3 a	5.7 ± 0.4 ab	3.66 ± 0.24 a	0.53 ± 0.15 a
	BTH High	13.3 ± 0.2 a	5.4 ± 0.2 b	3.82 ± 0.31 a	0.57 ± 0.02 a
Mouhtaro 2020	Control	14.2 ± 0.4 a	6.8 ± 0.5 a	3.8 ± 0.21 a	0.27 ± 0.03 a
	BTH Low	13.2 ± 0.3 b	6.1 ± 0.6 ab	3.8 ± 0.22 a	0.33 ± 0.03 a
	BTH High	13.4 ± 0.2 b	5.6 ± 0.1 b	3.9 ± 0.06 a	0.29 ± 0.02 a

Data represent mean ± std. deviation (*n* = 3). For each vintage and analysis, the mean values followed by different letters in the same column are significantly different according to the *t*-test at 5% probability.

**Table 7 plants-12-01179-t007:** Color and phenolic parameters of the three experimental Mouhtaro wines in response to two different BTH treatments.

Stage	Treatment	Total Phenolic Index	Color Intensity	Total Polyphenol Concentration (Gal. Ac. mg/L)	Total Anthocyanins (mg/L)	MCP
Mouhtaro 2019	Control	42.5 ± 2.5 b	13.0 ± 0.3 b	1506.5 ± 81.6 b	355.5 ± 4.5 b	232.3 ± 14.6 b
	BTH Low	46.0 ± 1.7 b	14.0 ± 0.5 a	1675.4 ± 60.2 ab	385.1 ± 12.7 a	234.5 ± 12.5 b
	BTH High	51.4 ± 2.2 a	13.3 ± 0.2 ab	1847.4 ± 133.1 a	363.9 ± 9.9 ab	339.7 ± 13.6 a
Mouhtaro 2020	Control	36.9 ± 3.7 b	12.5 ± 0.2 ab	1273.1 ± 49.3 c	325.5 ± 2.0 b	222.3 ± 6.6 c
	BTH Low	45.7 ± 3.7 a	13.0 ± 0.1 a	1442.1 ± 32.4 b	344.4 ± 8.1 a	237.8 ± 6.4 b
	BTH High	47.7 ± 3.4 a	12.2 ± 0.4 b	1610.7 ± 51.8 a	321.9 ± 3.8 b	323.1 ± 7.1 a

Data represent mean ± std. deviation (*n* = 3). For each vintage and analysis the mean values followed by different letters in the same column are significantly different according to *t*-test at 5% probability.

**Table 8 plants-12-01179-t008:** Individual anthocyanins evaluated by HPLC in Mouhtaro wines.

Vintage	Treatment	Dp3G	Cy3G	Pt3G	Pn3G	Mlv3G	MlvAc	MlvCm	Total Anthocyanins
Mouhtaro 2019	Control	32.0 ± 2.5 a	2.2 ± 0.1 b	41.1 ± 1.9 b	5.7 ± 0.5 b	260.0 ± 8.0 b	10.2 ± 1.0 a	4.7 ± 0.4 a	348.1 ± 7.9 b
	BTH Low	38.2 ± 4.1 a	2.6 ± 0.1 a	50.1 ± 2.7 a	7.8 ± 1.1 a	265.1 ± 1.9 a	8.4 ± 0.4 b	4.3 ± 0.6 ab	372.6 ± 4.8 a
	BTH High	38.7 ± 4.5 a	2.4 ± 0.1 a	49.5 ± 3.6 a	7.8 ± 1.1 a	251.1 ± 6.2 c	8.1 ± 0.4 b	3.5 ± 0.5 b	354.6 ± 8.6 b
Mouhtaro 2020	Control	27.0 ± 2.5 b	2.3 ± 0.1 ab	26.5 ± 1.6 b	2.6 ± 0.1 a	248.4 ± 3.1 a	12.6 ± 1.3 a	6.4 ± 0.8 a	319.6 ± 3.4 ab
	BTH Low	33.7 ± 1.2 a	2.4 ± 0.1 a	29.4 ± 0.6 a	3.1 ± 0.4 a	246.1 ± 1.4 ab	10.6 ± 1.2 a	5.5 ± 0.3 a	325.6 ± 0.9 a
	BTH High	34.2 ± 0.3 a	2.1 ± 0.1 b	25.8 ± 0.6 b	2.7 ± 0.1 a	241.0 ± 4.9 b	10.9 ± 0.1 a	5.4 ± 0.1 a	317.1 ± 3.9 b

Data represent mean ± std. deviation (*n* = 3). For each vintage and compound, the mean values followed by different letters in the same column are significantly different according to the *t*-test at 5% probability. Abbreviations: Dp3G. delphinidin-3-*O*-glucoside; Pt3G. petunidin-3-*O*-glucoside; Pn3G. peonidin-3-*O*-glucoside; Mlv3G. malvidin-3-*O*-glucoside; MlvAc. malvidin *3*-*O*-acetate-glucoside; MvCm. malvidin *3*-*O*-coumarate.

## Data Availability

The data presented in this study are available on request from the corresponding authors (pending privacy and ethical considerations).

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
