# Peer review of "Benzothiadiazole Affects Grape Polyphenol Metabolism and Wine Quality in Two Greek Cultivars: Effects during Ripening Period over Two Years"

_plants, 2023, doi:10.3390/plants12051179_

Round 1

Reviewer 1 Report

This paper addresses the effect of benzothiadiazole on polyphenol biosynthesis during grape ripening in Mouhtaro (red-colored) and Savvatiano (white-colored) greek cultivars. It is a very interesting and novel work that addresses the effect of an elicitor that can improve aspects of quality in berries and wines. The work is well structured, presenting the results in an orderly and clear manner, with discussion in relation to other authors and with conclusions that respond to the objectives set. My opinion is that it should be published with minimal modifications.

Here are some aspects that need to be modified:

- Change in both text (lines 221, 359, and tables (4, 7), "Folin-Ciocalteau's" for the "total polyphenol concentration", as it appears in the Materials and Methods.

- When HPLC is indicated (lines 379, 394; Table 8), the type of detector is not mentioned and neither in the materials and methods details of the equipment and methodology are given.

- In the line 388, change malvidin (that is an anthocyanidin) by malvidin-3-O-glucoside.

- Lines 321-323. It is normal and expected that the expression of UDP-glucose-flavonoid 3-O-glycosyltransferase gene (VviUFGT) is higher in a red variety than in a white one, but it is strange that there is expression of VviUFGT in a white variety (which does not present anthocyanins), so it would be advisable to explain these results in more detail.

Author Response

Answers to Reviewer 1

This paper addresses the effect of benzothiadiazole on polyphenol biosynthesis during grape ripening in Mouhtaro (red-colored) and Savvatiano (white-colored) greek cultivars. It is a very interesting and novel work that addresses the effect of an elicitor that can improve aspects of quality in berries and wines. The work is well structured, presenting the results in an orderly and clear manner, with discussion in relation to other authors and with conclusions that respond to the objectives set. My opinion is that it should be published with minimal modifications.

Authors would like to thank reviewer for the comments. 

Here are some aspects that need to be modified:

- Change in both text (lines 221, 359, and tables (4, 7), "Folin-Ciocalteau's" for the "total polyphenol concentration", as it appears in the Materials and Methods.

We agree with the comment. Changes have been adapted at the manuscript at Table 4 and Line 389 of the new version of the manuscript.

- When HPLC is indicated (lines 379, 394; Table 8), the type of detector is not mentioned and neither in the materials and methods details of the equipment and methodology are given.

We added the references in which the method is described in details. However, as the reviewer mentioned some info are missing. Hence we added in the field of Materials and Methods and more especially in the paragraph “4.6.4 Anthocyanins by HPLC” the methodology followed in details.

In details, reversed–phase HPLC analyses of anthocyanins were carried out by direct injection of 10 μL of wine into a Waters 2695 Alliance liquid chromatograph system coupled with a Waters 2996 PDA detector (Milford, MA, USA) and using a SVEA C18 Plus 4.6 × 250 mm, 5 μm column (Nanologica, Södertälje, Sweden). The mobile phases used were 10% aqueous formic acid (solvent A) and methanol (solvent B). Chromatograms were recorded at 520 nm, and anthocyanin standard curves were made using malvidin–3–O–glucoside chloride. Identification was based on comparing retention times of the peaks detected with those of original compounds, and on UV–Vis spectrum. The anthocyanidin–3–O-monoglucosides delphinidin, peonidin, petunidin, and malvidin and the acetylated and p–coumarylated of malvidin were expressed as mg/L of malvidin–3–O–glucoside.- In the line 388, change malvidin (that is an anthocyanidin) by malvidin-3-O-glucoside.

- Lines 321-323. It is normal and expected that the expression of UDP-glucose-flavonoid 3-O-glycosyltransferase gene (VviUFGT) is higher in a red variety than in a white one, but it is strange that there is expression of VviUFGT in a white variety (which does not present anthocyanins), so it would be advisable to explain these results in more detail.

According to the reviewer’ s comment the following paragraph were added in the manuscript:

At the manuscript’ s paragraph  “2 Metabolomic Analysis of Savvatiano grape berries in response to BTH”

Conventional HPLC methods which are used in order to detect and quantify grape anthocyanin are not sensitive enough to detect pigments at the level of a few μg/kg grapes. Although, modern UPLC-MS/MS instrument which are characterised by a higher number of chromatographic theoretical plates and a higher sensitivity detector (triple quadruple MS), is also able to detect and quantify traces of anthocyanins. So far, few research studies have demonstrated the existence of anthocyanins in white grapevine varieties such as in Vitis vinifera L. Siria. (Andrea Silva et al., 2014) Chardonnay, Riesling and Sauvingnon Blanc (Arapitsas et al 2015).

At the manuscript’ s paragraph  “2.11. Gene Expression”

In the final step of anthocyanin synthesis, all the genes of the flavonoid pathway are present both in white and red grape berries, including a UDP glucose-flavonoid 3-O-glucosyl transferase (UFGT) (Boss et al., 1996; Boss et al 1996b). Moreover, Walker et al. (2007) found that two very similar regulatory genes, VvMYBA1 and VvMYBA2, which could activate anthocyanin biosynthesis were not transcribed in white skin berries. (Castellarin and Di Gaspero, 2007). Nevertheless, the existence of several other MYB-type transcription factors that can modulate flavonoid biosynthesis (Ferreira et al., 2019) and the identification of VviUFGT in transcriptomic studies in white-colored cultivars (Savoi et al., 2016), imply more complicated  regulatory mechanisms.

 Arapitsas P, Oliveira J, Mattivi F. Do white grapes really exist? Food Research International. 2015;69:21-25. doi:10.1016/j.foodres.2014.12.002

Andrea-Silva,J.,Cosme,F.,Ribeiro,L.F.,Moreira,A.S.P.,Malheiro,A.C.,Coimbra,M.A.,etal. (2014). Origin of the pinking phenomenon of white wines. Journal of Agricultural and Food Chemistry, 62(24), 5651–5659, http://dx.doi.org/10.1021/jf500825h.

Boss, P. K., Davies, C., & Robinson, S. P. (1996a). Expression of anthocyanin biosynthesis pathway genes in red and white grapes. Plant Molecular Biology, 32(3), 565–569.

Boss, P. K., Davies, C., & Robinson, S. P. (1996b). Anthocyanin composition and anthocyanin pathway gene expression in grapevine sports differing in berry skin colour. Australian Journal of Grape and Wine Research, 2(3), 163–170, http://dx.doi.org/10. 1111/j.1755-0238.1996.tb00104.x.

Castellarin, S. D., & Di Gaspero, G. (2007). Transcriptional control of anthocyanin biosynthetic genes in extreme phenotypes for berry pigmentation of naturally occurring grapevines. BMC Plant Biology, 7(1), 46, http://dx.doi.org/10.1186/1471-2229-7-46

Walker, A. R., Lee, E., Bogs, J., McDavid, D. A. J., Thomas, M. R., & Robinson, S. P. (2007). White grapes arose through the mutation of two similar and adjacent regulatory genes. The Plant journal: for cell and molecular biology, 49(5), 772–785, http://dx.doi. org/10.1111/j.1365-313X.2006.02997.x.

Ferreira, V.; Matus, J.T.; Pinto-Carnide, O.; Carrasco, D.; Arroyo-García, R.; Castro, I. Genetic Analysis of a White-to-Red Berry Skin Color Reversion and Its Transcriptomic and Metabolic Consequences in Grapevine (Vitis Vinifera Cv. ‘Moscatel Galego’). BMC Genom. 2019, 20, 952.

Savoi, S.; Wong, D.C.J.; Arapitsas, P.; Miculan, M.; Bucchetti, B.; Peterlunger, E.; Fait, A.; Mattivi, F.; Castellarin, S.D. Transcriptome and metabolite profiling reveals that prolonged drought modulates the phenylpropanoid and terpenoid pathway in white grapes (Vitis vinifera L.). BMC Plant Biol. 2016, 16, 67.

Moreover, several spelling and grammar revision were done in the manuscript by an Australian translator of the English language

Reviewer 2 Report

Dear Authors,

The experiment was conducted in two growing seasons (2019–2020) to investigate the effect of benzothiadiazole on polyphenol biosynthesis during grape ripening in Mouhtaro and Savvatiano varieties. Grapevines were treated at the stage of veraison with 0.3 mM and 0.6 mM benzothiadiazole. The phenolic content of grapes, as well as the expression level of genes involved in the phenylpropanoid pathway, were evaluated and showed an induction of genes specifically engaged in anthocyanins and stilbenoids biosynthesis.

- line 50 add the following sentence "In the last decade, research groups around the world studied the beneficial application of biostimulants on different grapevine varieties [9-15].

[14] Cataldo, E., Fucile, M., & Mattii, G. B. (2022). Leaf Eco-Physiological Profile and Berries Technological Traits on Potted Vitis vinifera L. cv Pinot Noir Subordinated to Zeolite Treatments under Drought Stress. Plants, 11(13), 1735.

[15] Cataldo, E., Fucile, M., Manzi, D., Masini, C. M., Doni, S., & Mattii, G. B. (2023). Sustainable Soil Management: Effects of Clinoptilolite and Organic Compost Soil Application on Eco-Physiology, Quercitin, and Hydroxylated, Methoxylated Anthocyanins on Vitis vinifera. Plants, 12(4), 708.

- Authors should implement the introduction with more information about benzothiadiazole (BTH) (chemical information, applications, and case studies)

The work is interesting and well-conducted.

A revision in language and punctuation is suggested

Author Response

Answer to Reviewer 2

The experiment was conducted in two growing seasons (2019–2020) to investigate the effect of benzothiadiazole on polyphenol biosynthesis during grape ripening in Mouhtaro and Savvatiano varieties. Grapevines were treated at the stage of veraison with 0.3 mM and 0.6 mM benzothiadiazole. The phenolic content of grapes, as well as the expression level of genes involved in the phenylpropanoid pathway, were evaluated and showed an induction of genes specifically engaged in anthocyanins and stilbenoids biosynthesis.

Authors would like to thank reviewer for the comments. 

- line 50 add the following sentence "In the last decade, research groups around the world studied the beneficial application of biostimulants on different grapevine varieties [9-15].

Cataldo, E., Fucile, M., & Mattii, G. B. (2022). Leaf Eco-Physiological Profile and Berries Technological Traits on Potted Vitis vinifera L. cv Pinot Noir Subordinated to Zeolite Treatments under Drought Stress. Plants, 11(13), 1735.

 Cataldo, E., Fucile, M., Manzi, D., Masini, C. M., Doni, S., & Mattii, G. B. (2023). Sustainable Soil Management: Effects of Clinoptilolite and Organic Compost Soil Application on Eco-Physiology, Quercitin, and Hydroxylated, Methoxylated Anthocyanins on Vitis vinifera. Plants, 12(4), 708.

The current research work is more oriented on the bioestimulats/elicitors (benzothiadiazole) applied at the vine and how it induces the biosynthesis of primary and secondary metabolites. It less related with applications at the vine or even the soil with indirect impact on grape physiology. They are a lot and interesting research works in this this topic, but the scopus of the present manuscript and the limitation to mention all of them make it impossible to include all those works. Therefore, no changes have been made.

- Authors should implement the introduction with more information about benzothiadiazole (BTH) (chemical information, applications, and case studies)

Chemical information about BTH are given in the part of Materials and Methods, we added some more information about BTH and its use. However, due to the fact that the current research is more oriented in the oeno- viticultural field we provide more information about the application of BTH in international grapevine varieties and its impact the quality characteristics of grapes and the produced wines.

Therefore the following paragraph added and the updated bibliography as well

In the last 10 years, several studies have been conducted regarding  the impact of the biostimulants. The exogenous applications of biostimulants induce the activation of enzymes involved in the biosynthesis of phenolic compounds (Salifu et la., 2022; Ge et al., 2019; Bektas and Euglen, 2015). BTH is considered to be an analogue of Salicilic Acid inducing resistance against a broad spectrum of plant–pathogen (Friedrich et al., 1996; Lawton et al., 1996; Jiang et al., 2015)

Salifu, R.; Chen, C.; Sam, F.E.; Jiang, Y. Application of Elicitors in Grapevine Defense: Impact on Volatile Compounds. Horticulturae 2022, 8, 451.

Ge, Y.; Tang, Q.; Li, C.; Duan, B.; Li, X.; Wei, M.; Li, J. Acibenzolar-S-methyl treatment enhances antioxidant ability and phenylpropanoid pathway of blueberries during low temperature storage. LWT-Food Sci. Technol. 2019, 110, 48–53.

Bektas, Y.; Eulgem, T. Synthetic plant defense elicitors. Front. Plant Sci. 2015, 5, 804.

Friedrich L, Lawton K, Ruess W, Masner P, Specker N, GutRella M et al., A benzothiadiazole derivative induces systemic acquired resistance in tobacco. Plant J 10:61–70 (1996).

Lawton K, Friedrich L, Hunt M, Weymann K, Delaney T, Kessmann H et al., Benzothiadiazole induces disease resistance in Arabidopsis by activation of the systemic acquired resistance signal transduction pathway. Plant J 10:71–82 (1996).

Jiang, H.; Wang, Y.; Li, C.; Wang, B.; Ma, L.; Ren, Y.; Bi, Y.; Li, Y.; Xue, H.; Prusky, D. The effect of benzo-(1,2,3)-thiadiazole-7carbothioic acid S-methyl ester (BTH) treatment on regulation of reactive oxygen species metabolism involved in wound healing of potato tubers during postharvest. Food Chem. 2020, 309, 125608.

The work is interesting and well-conducted.

Authors thanks reviewer for his positive feedback.

A revision in language and punctuation is suggested

Manuscript was reviewed from a native Australian translator of the English language.
